# Are We Benign? What Can Wnt Signaling Pathway and Epithelial to Mesenchymal Transition Tell Us about Intracranial Meningioma Progression

**DOI:** 10.3390/cancers13071633

**Published:** 2021-04-01

**Authors:** Anja Bukovac, Anja Kafka, Marina Raguž, Petar Brlek, Katarina Dragičević, Danko Müller, Nives Pećina-Šlaus

**Affiliations:** 1Laboratory of Neurooncology, Croatian Institute for Brain Research, School of Medicine, University of Zagreb, 10000 Zagreb, Croatia; anja.bukovac@mef.hr (A.B.); anja.kafka@mef.hr (A.K.); pbrlek@gmail.com (P.B.); dragicevic1996@gmail.com (K.D.); 2Department of Biology, School of Medicine, University of Zagreb, 10000 Zagreb, Croatia; 3Department of Neurosurgery, University hospital Dubrava, 10000 Zagreb, Croatia; marinaraguz@gmail.com; 4Department of Pathology and Cytology, University Hospital Dubrava, 10000 Zagreb, Croatia; danko.mueller@yahoo.com

**Keywords:** epithelial to mesenchymal transition (EMT), Wnt signaling pathway, E-cadherin, N-cadherin, TWIST1, SNAIL and SLUG, β-catenin, intracranial meningioma

## Abstract

**Simple Summary:**

Intracranial meningiomas are one of the most common primary brain tumors. Although mostly benign, a small portion may exhibit aggressive and malignant characteristics leading to higher recurrence and mortality rate. Detecting the molecular profiles and genes that are involved in meningioma progression can lead to better stratification of patients and more efficient and targeted treatments. The results of this study reveal the role of main actors of the Wnt signaling pathway (β-catenin) and epithelial to mesenchymal transition (E-cadherin, N-cadherin, TWIST1, SNAIL and SLUG) in progression of these tumors, potentially bringing novel biomarkers in diagnostics and molecular targets for therapeutic interventions.

**Abstract:**

Epithelial to mesenchymal transition (EMT), which is characterized by the reduced expression of E-cadherin and increased expression of N-cadherin, plays an important role in the tumor invasion and metastasis. Classical Wnt signaling pathway has a tight link with EMT and it has been shown that nuclear translocation of β-catenin can induce EMT. This research has showed that genes that are involved in cadherin switch, *CDH1* and *CDH2*, play a role in meningioma progression. Increased N-cadherin expression in relation to E-cadherin was recorded. In meningioma, transcription factors SNAIL, SLUG, and TWIST1 demonstrated strong expression in relation to E- and N-cadherin. The expression of SNAIL and SLUG was significantly associated with higher grades (*p* = 0.001), indicating their role in meningioma progression. Higher grades also recorded an increased expression of total β-catenin followed by an increased expression of its active form (*p* = 0.000). This research brings the results of genetic and protein analyzes of important molecules that are involved in Wnt and EMT signaling pathways and reveals their role in intracranial meningioma. The results of this study offer guidelines and new markers of progression for future research and reveal new molecular targets of therapeutic interventions.

## 1. Introduction

Meningiomas are one of the most common primary tumors of central nervous system that the World Health Organization (WHO) (2016) defines as: “A group of mostly benign, slow-growing neoplasms that most likely derive from the meningothelial cells of the arachnoid layer”. Although mostly benign (about 80% of cases), some meningioma classified as grades II (atypical) and grades III (anaplastic), show aggressive behavior and the latest one malignant characteristic. Patients that are diagnosed with anaplastic meningioma could potentially develop metastasis leading to poor survival prognosis [1,2,3]. Also, tumors of the same pathohistological type do not always have the same outcome or may respond differently to therapies based on their differences in molecular pathways or genetic profiles. Today, the molecular profile of tumors has become crucial in the diagnosis and choice of therapy, which is the reason the WHO has included it in its classification from 2016 [2]. The malfunction of signaling pathways in cancer cells is often responsible for their resistance to various forms of treatment, primarily chemotherapy. Genetic changes and genomic instability are the main features of tumors, because they often result in defective and/or non-functional protein products that disrupt cell function [4]. Genomic instability comes in various forms, the most common of which are chromosomal instability and microsatellite instability (MSI) because of the malfunction of the DNA mismatch repair (MMR) system [5,6]. The recognition of these events in certain types of tumors is of great importance due to the discovery of biomarkers that can serve to determine targeted and personalized therapy. To improve diagnostic, predict behavior and development of meningioma, which can result in precise and personal treatment, a search for valid prognostic biomarkers is in progress.

The Wnt signaling pathway is one of the basic cellular pathways essential during the embryonic development and it is suspected to have a role in meningioma tumorigenesis. The main actor of canonical form of Wnt signaling pathway is β-catenin. In inactive form of Wnt signaling pathway, β-catenin that accumulates in the cytoplasm is targeted and phosphorylated by a destruction complex made of AXIN, APC, GSK3β, and CK1. In the active form of signaling pathway the destruction complex is attracted to the cell membrane and β-catenin is not phosphorylated, but rather it stabilizes in the cytoplasm and, consequently, passes into the nucleus. In the nucleus, together with transcription factors TCF/LEF, β-catenin participates in the transcription of target genes. It has been shown that over 90% of cancer-related β-catenin mutations are mutations in its exon 3 of the *CTNNB1* gene [7]. Mutations in this exon lead to the inhibition of β-catenin phosphorylation, thus enabling its stabilization and consequently possible activation of the Wnt signaling pathway.

Epithelial-to-mesenchymal transition (EMT) is a biological process that is necessary for embryogenesis. During EMT, cells undergo molecular changes and become motile. The most prominent feature of EMT is the so-called cadherin switch, in which the partial or complete loss of the expression of E-cadherin and, respectively, epithelial phenotype is substituted with the increased expression of N-cadherin and the acquisition of mesenchymal phenotype. That means that, in tumorigenesis, EMT could lead to the development of malignancy and metastasis. Today, it is known that successful tumor invasion is more often the result of partial EMT characterized by the simultaneous expression of markers of epithelial and mesenchymal characteristics or loss of epithelial characteristics that are not replaced by mesenchymal [8,9,10]. Thanks to partial EMT, tumors develop plasticity that allows them to adapt to the new microenvironment, leading to the development of a secondary tumor through revers process called mesenchymal to epithelial transition (MET). Partial EMT can also be the cause of treatment ineffectiveness and relapse [11,12]. EMT is regulated at several levels, the most pronounced of which is the level of transcription. Transcription factors from the family of proteins SNAIL (SNAI1, SNAI2/SLUG, SNAI3/SMUC), ZEB (ZEB1/TCF8, ZEB2/SIP1), TWIST (TWIST1, TWIST2), FOXCs (FOXC1, FOXC2), and TCF12 (EO) suppress the expression of E-cadherin (*CDH1*) by binding to its promoter site and stopping its transcription [10,13,14,15,16].

The classical Wnt pathway has a tight link with EMT, and it has been shown that the nuclear translocation of β-catenin can induce EMT. Bound to E-cadherin in adherens junctions is β-catenin, whose translocation to the nucleus is yet another molecular event that is involved in EMT. The stabilization and nuclear accumulation of beta-catenin can activate the transcriptional repressors SNAIL and SLUG that suppress E-cadherin expression and, thus, induce EMT.

The aim of our study was to investigate the involvement of main actors of the Wnt signaling pathway (β-catenin) and EMT (E-cadherin, N-cadherin, TWIST1, SNAIL and SLUG) in the progression of meningioma to its invasive grade.

## 2. Materials and Methods

### 2.1. Tissue Collection

Tumor tissue samples of intracranial meningioma were collected from 72 patients of different sex and age who were volunteers and have already been scheduled for surgery. Tumor tissues were not treated with other oncological methods (radiation, chemotherapy) prior to their sampling. All of the tumors were classified, and grades and subtypes were assigned by a pathologist. Classification was carried out according to WHO guidelines [2]. Our sample consisted of 49 benign, 17 atypical, and six anaplastic meningiomas. Along with the tumor tissue, 3–5 mL of blood was collected from each patient and then used as the control in genetic analysis. The study was approved by the Ethics Committee of School of Medicine University of Zagreb (Case number: 380-59-10106-17-100/98; Class: 641-01/17-02/01, 23 March 2017), Ethics Committee of University Hospital Center Zagreb (number 02/21/AG, class: 8.1-16/215-2, 2 February 2017), Ethics Committee of University Hospital Center “Sisters of Charity” (number EP-5429/17-5, 23 March 2017) and the Ethics Committee of University hospital Dubrava (17 May 2017).

### 2.2. DNA Extraction and Analysis

The isolation of DNA from tumor tissue was performed with standard phenol/chloroform method [17], while isolation of DNA from blood was performed with standard salting out method [18]. 

For analyzes of the genetic changes of E-cadherin and N-cadherin, their *CDH1* and *CDH2* genes were amplified using a set of known and highly polymorphic microsatellite markers that were located near or within the gene itself. Microsatellite markers were chosen by searching the publicly available human genome builds and they are available in the human gene bank Ensemble (http://www.ensembl.org, accessed on 20 September 2018) and NCBI (https://www.ncbi.nlm.nih.gov, accessed on 20 September 2018). Microsatellite markers with a high percentage of heterozygosity in the population were selected for gene analyzes, because they give the so-called informative or heterozygous individuals with different numbers of allele repetitions. To investigate changes in the *CDH1* gene, microsatellite markers D16S752 and D16S3025 were used, while microsatellite markers D18S66 and D18S819 were used for the *CDH2* gene. Table 1 shows the PCR conditions for these markers. Genetic changes—loss of heterozygosity (LOH) or microsatellite instability (MSI)—were analyzed by electrophoresis on high-resolution Spreadex gels (EL400 Mini and EL600 Mini, Elchrom Scientific, AL-Diagnostic GmbH, Austria). To detect genetic changes, DNA from tumor and DNA from the blood of the same patient were compared. The loss of band in the tumor indicated LOH, while the appearance of additional bands or bands at different positions in the tumor relative to the blood indicated MSI.

The *CTNNB1* gene that encodes β-catenin was inspected for mutation in its exon 3. A quick search for mutations in exon 3 of *CTNNB1* gene was performed by high-resolution melting analysis (HRM) on a Roche Light Cycler^®^ Nano System device (Roche, Basel, Switzerland) that combines PCR and HRM methods into one successive procedure. Each HRM analysis was preceded by the amplification of the gene of interest from tumor DNA and blood DNA of patients. The melting curves were analyzed in LightCycler^®^ Nano 1.1 software. If the melting curves suggested the presence of mutation, those samples were amplified by PCR using custom made primers – 5′ CCAATCTACTAATGCTAATACTG 3′ and 5′ CTGCATTCTGACTTTCAGTAAGG 3′, which cover whole exon 3 and part of intron 3 (Table 1). The amplicons were then purified with FastAP kit (ThermoFisher Scientific, Waltham, MA, USA) and sequenced using standard Sanger sequencing with BigDyeTerminator v3.1 Cycle Sequencing kit on ABI 3730XL (Applied Biosystems, Foster City, CA, USA). Mutation detection was performed by comparing the DNA sequences from the tumor with the blood sequences of the same patient and the publicly available exon 3 β-catenin sequence that was taken from the NCBI database (https://www.ncbi.nlm.nih.gov/nuccore/NG_013302.2?&feature%20=%20any, accessed on 10 November 2019). 

An evaluation of the effect of found mutations in exon 3 of the *CTNNB1* gene on its 76 amino acids protein product was performed using the program “Porter 5.0: Prediction of protein secondary structure” (http://distilldeep.ucd.ie/porter/, accessed on 12 March 2020) [19].

### 2.3. Protein Localization and Expression

Protein localizations and expressions were analyzed for β-catenin, E-cadherin, N-cadherin, TWIST1, SNAIL, and SLUG. For this purpose, immunohistochemistry using Peroxidase/DAB (EnVision^TM^, Dako REAL^TM^, Glostrup, Denmark) was performed on 4-μm-thick paraffin embedded sections that were fixed onto microscope slides (DakoCytomation, Glostrup, Denmark). To prepare samples for staining, we deparaffinized sections in xylene/ethanol baths and rehydrated in citrate buffer (C_6_H_8_O_7_·H_2_O; pH 6,0), which unmasked epitopes. To avoid false staining, we fixed sections with 3% H_2_O_2_ for 10 min in dark chamber and incubated sections with goat serum for 30 min at 4 °C. Afterwards, the tissue was incubated with an optimally diluted primary antibody over night at 4 °C, except for negative controls. In this research, we used five monoclonal antibodies and one polyclonal antibody. For E-cadherin detection, we used monoclonal E-cadherin clone: NCH-38 Code M3612 (Dako), diluted 1:100; for N-cadherin detection monoclonal antibody raised against amino acids 450–512 within the extracellular domain of N-cadherin of human origin (N-cadherin (D-4): sc-8424 (Santa Cruz Biotechnology, Inc., Dallas, TX, USA), diluted 1:200; for TWIST1 detection monoclonal Anti-Twist antibody [10E4E6] ab175430 (Abcam, Cambridge, UK), diluted 1:400; for SNAIL and SLUG detection polyclonal Anti-SNAIL + SLUG antibody ab180714 (Abcam), and diluted 1:200. For β-catenin, we used two different antibodies for the detection of its phosphorylation status: non-Phospho beta-catenin (Ser33-37/Thr41) (D131A1) Rabbit mAb #8814 (Cell Signalling Technology), diluted 1:800 and monoclonal antibody beta-catenin clone b-Catenin-1 Code M3539 (Dako), and diluted 1:200. The following step was incubation with Dako REAL™EnVision™/HRP, Rabbit/Mouse (ENV) reagent for 45 min and visualization of reaction by Dako REAL™ DAB+ Chromogen (EnVisionTM, Dako REALTM) for 1–5 min. The counterstaining with hematoxylin histological staining reagent (Dako) was applied for 3 min. 

The results of the immunohistochemistry were analyzed with Olympus BH-2 and IX83 light microscopes and then scanned using a digital scanner (NanoZoomer 2.0 RS: Hamamatsu Photonics, Hamamatsu City, Japan). 

To evaluate immunopositivity of each sample, a minimal number of 200 cells were counted in the tumor hotspot. Cells with specific expression level were counted using ImageJ software (National Institutes of Health, Bethesda, MD, USA) and then evaluated by a semiquantitative method. The quantification was done with H-score (Histo-score), which includes the sum of individual H-scores for each intensity and gives a higher value to higher intensities of protein expression:H = [1 × (% cells 1+) + 2 × (% cells 2+) + 3 × (% cells 3+)]
where 1+ indicates weak immunopositivity—yellowish/light brown color, 2+ indicates moderate immunopositivity—light brown, and 3+ indicates strong immunopositivity—dark brown. By calculating the H-score, a range of protein expression values on a scale of 0–300 was obtained [20]. Based on the H-score value, the protein expression of the samples was categorized into three categories of signal strength: 0–100 = no signal/weak signal (0/1+), 101–200 = moderate signal (2+), and 201–300 = strong signal (3+).

### 2.4. Statistical Analysis

Statistical analysis of the results that were obtained by genetic and protein analyses was done in the software package SPSS v.19.0.1 (SPSS, Chicago, IL, USA), with a selected significance level of *p* ≤ 0.05. All of the variables obtained by genetic and protein analyzes were correlated with variables of collected pathohistological and demographic (sex, age) parameters. Before selecting appropriate statistical tests, the distribution of normality of the included variables was obtained with the Kolmogorov–Smirnov test. When testing the significance of the difference between the two groups, the t-test for independent samples was used (or Mann–Whitney *U* test as its nonparametric replacement), when testing the significance of the difference between more than two groups, one-way analysis of variance (ANOVA) was used (or Kruskal–Wallis test as nonparametric replacement). To test the correlation in the distributions of two or more variables, the χ^2^ test and the corresponding correlation coefficients (Φ coefficient in the case of 2 × 2 tables or Cramer’s V coefficient in the case of asymmetric tables) were used. The Pearson r correlation coefficient was used to test the correlation between two continuous variables or one continuous and one naturally dichotomous variable.

## 3. Results

### 3.1. Loss of Heterozygosity and Microsatellite Instability of CDH1 and CDH2 Gene

In a total sample of 72 intracranial meningioma, both markers for *CDH1* proved to be highly informative. Marker D16S752 showed heterozygosity in 95.8% of cases, while marker D16S3025 in 90.3% of cases. A pooled analysis of the two markers for the gene *CDH1* showed that each sample was informative in at least one marker. Microsatellite instability was observed in 14 samples (19.4%), loss of heterozygosity in nine samples (12.5%), and two samples (2.8%) harbored both genetic changes. Further analysis showed that the anaplastic meningioma group contained the highest percentage of MSI and LOH (Figure 1a and Figure 2). Statistical analysis did not show a significant difference between the occurrence of pooled analysis of genetic changes with demographic variables (sex *p* = 0.960 and age *p* = 0.584) or grade (*p* = 0.126).

The markers for *CDH2* gene D18S66 and D18S819 were found to be highly informative with 100% heterozygosity in a total sample of 70 intracranial meningioma. A pooled analysis of the two markers of the gene *CDH2* revealed a great number of genetic changes (Figure 1b and Figure 2) across all grades. 70% of the samples showed one or more genetic changes, 28 samples (40%) microsatellite instability, eight samples (11.4%) loss of heterozygosity, and 13 samples (18.6%) both types of genetic changes. The pooled analysis of *CDH2* genetic changes also did not show a statistically significant association of the frequency of these changes with sex (*p* = 0.620), age (*p* = 0.151), or grade (*p* = 0.307). However, it is notable from Figure 1 that anaplastic cases harbored the highest percentage of MSI and LOH.

The results of genetic analyzes with microsatellite markers for the *CDH1* and *CDH2* genes were compared to see whether their mutual occurrence was related. The investigation of these two genes relationship revealed a statistically significant association between the occurrence of genetic changes between markers D16S725 and D18S66 (φc = 0.294; *p* = 0.016) and between markers D16S3025 and D18S66 (φc = 0.489; *p* = 0.000). Additionally, association was found in cases when both microsatellite markers of the *CDH1* gene were compared with the *CDH2* gene marker D18S66 (φc = 0.307; *p* = 0.010) or when both microsatellite markers of the *CDH2* gene were compared with the *CDH1* gene marker D16S3025 (φc = 0.353; *p* = 0.005). These results suggest that, in our sample of intracranial meningioma, if the *CDH1* gene shows a genetic change, it is very likely that the *CDH2* gene will also be altered.

### 3.2. Mutations in Exon 3 β-Catenin (CTNNB1) and Prediction of Their Effects

A total number of 63 samples of intracranial meningioma and autologous blood were available for HRM analysis and standard Sanger sequencing. The results showed that 14 samples (22.2%) of intracranial meningioma harbored mutation in β-catenin. Mutations were recorded in exon 3, as well as in part of the intron 3, which was covered during sequencing (Figure 3). Of the total number of samples with mutations, three samples showed mutations in the exon, eight in the intron, and three samples had mutations in both the exon and the intron. The samples with mutation in the exon were further analyzed to determine their effect on the 76 amino acids long protein product with “Porter 5.0: Prediction of protein secondary structure” program. Simulation and prediction of secondary protein structure showed that mutations in the exon 3 of five samples cause earlier cessation of protein synthesis because of premature stop codon in the sequence, which results in shorter proteins that are often dysfunctional. Simulation of the secondary protein structure of the remaining sample mutated in exon 3 showed that base substitution in the exon did not cause a difference in the structure of the formed protein, however two intron insertions in the same sample could cause the shift of reading frame and incorrect intron excision, again leading to protein dysfunction. Because of the above facts, all mutated samples contain at least one genetic change that affects protein functionality and is potentially harmful. Samples with mutations in exon 3 did not show the changes that most commonly occur in S33, S37, T41, and S45, but they occurred at other locations. In 3/6 cases of mutations in exon 3, protein synthesis was stopped before S33, S37, T41, S45, and either due to a shift in the reading frame or due to a missense mutation, the synthesis of these amino acids did not occur at these locations. In 2/6 of the cases, protein synthesis was stopped after S33, S37, T41, and S45, whereby the mentioned amino acids were formed, but the whole N-terminal end of β-catenin was shorter. Mutations in intron 3 occurred in 11/12 samples at two locations in the genome with insertion of A base (g.30340_30341insA and g.30343_30344insA).

A total of 14 insertions, four deletions, and four substitutions were detected, which resulted in 18 frame shifts, two nonsense mutations, one missense mutation and one silent mutation. In most samples (13 of 14), the signal of the non-phosphorylated (active) form of β-catenin was equal to the signal of total β-catenin. In meningioma with mutations that cause shorter nonfunctional proteins (premature termination of protein synthesis), immunohistochemical analysis found that three samples (60%) had a weak signal of both studied forms of β-catenin (non-phosphorylated/active and total), while one sample had a weak signal of the non-phosphorylated form of β-catenin and a moderate signal of total β-catenin, and one sample had a moderate signal of both forms of β-catenin. The silent mutation that is accompanied by potentially crooked intron excision had a weak signal of the non-phosphorylated form of β-catenin and a strong signal of total β-catenin. Of the total number of meningioma with potentially wrong intron excision, four samples (50%) showed a weak signal of both forms of β-catenin, while three samples (37.5%) had a moderate signal of both forms of β-catenin, and one sample showed a strong signal of the non-phosphorylated form of β-catenin and a moderate signal of total β-catenin. Atypical meningiomas that harbored a mutation in the exon or intron 3 of *CTNNB1* showed a moderate signal of both forms of β-catenin in 75% of cases. 

The occurrence of the mutation was not statistically related to age (*p* = 0.073) or specific grade (*p* = 0.538). Mutations were observed in ten samples of benign meningioma, four samples of atypical meningioma, and they were not recorded in anaplastic samples.

### 3.3. Expression of EMT Markers—E-Cadherin and N-Cadherin

H-score analysis of the protein expression of E-cadherin on collected intracranial meningioma showed generally a very low incidence of protein. E-cadherin was recorded in the cytoplasm and membrane. The largest percentage of samples (77.8%) showed a weak signal, in which 22 samples (30.6%) lack signal. A moderate signal was observed in 19.4% of samples and only two samples (2.8%) showed values that were greater than 200. The mean H-score expression value was 48.83. A lack of expression was not significantly associated with grade (*p* = 0.116), and most of the samples showed a weak signal regardless of grade (81.6% benign, 70.6% atypical, and 66.7% anaplastic samples). While comparing the results of the E-cadherin H-score analysis with gender or age groups, no statistically significant correlation was observed (respectively: *p* = 0.513 and *p* = 0.816). Also, the expression of E-cadherin was not associated with the appearance of genetic changes recorded with microsatellite markers D16S752 (*p* = 0.236) and D16S3025 (*p* = 0.127) for the *CDH1* gene or with the appearance of genetic changes that were recorded with the marker D18S66 (*p* = 0.486) and D18S819 (*p* = 0.975) for the *CDH2* gene. Out of the 56 samples that had weak E-cadherin signal, 35.7% suffered change in the *CDH1* gene. The *CDH1* gene was changed in 28.6% of cases with a moderate signal while only one of two samples with a strong signal showed a genetic change in this gene. There was no significant correlation (*p* = 0.771) between the expression of E-cadherin and the occurrence of mutations in the *CTNNB1* gene. However, 10 out of 14 samples with *CTNNB1* mutations lacked or had low expression of E-cadherin.

The protein expression of N-cadherin was significantly more pronounced in intracranial meningioma as compared to E-cadherin expression. The mean value of expression that was obtained by H-score was 81.04. In most samples, a weak signal (54.2%) was recorded, including four samples (5.6%) that lacked signal. A moderate signal was noted in 27 (37.4%) samples, and 6 (8.3%) samples showed strong signal. The correlation of H-score value with grade, sex and age group was not statistically significant (respectively: *p* = 0.552, *p* = 0.511 and *p* = 0.537). However, a statistically significant difference was discovered in N-cadherin expression with respect to the presence of genetic changes in the *CDH1* (E-cadherin) gene revealed with marker D16S752 or D16S3025 (t = 2.100, df = 67, *p* = 0.040 and t = 2.589, df = 63, *p* = 0.012). The presence of *CDH1* genetic changes (revealed with both microsatellite markers) also showed a statistically significant difference in N-cadherin expression (t = 2.462, df = 70, *p* = 0.016). A more detailed *post-hoc* analysis found that the samples with the changes on E-cadherin gene had, on average, significantly lower H-score expression of N-cadherin. The expression of N-cadherin was not statistically significantly associated with changes that were recorded by the *CDH2* gene markers D18S66 (*p* = 0.272) and D18S819 (*p* = 0.181). However, of the 38 samples that showed poor N-cadherin expression, 73.7% had a change in the *CDH2* gene. Samples with moderate expression of N-cadherin showed almost identical results, where 73.1% of them had genetic change in the *CDH2* gene. Samples with a strong signal had fewer genetic changes, and they were recorded in 33.3% of the samples. No statistically significant association was found between N-cadherin expression and mutations in the *CTNNB1* gene (*p* = 0.098).

### 3.4. Expression of EMT Transcription Factors—TWIST1, SNAIL and SLUG

TWIST1 expression was exclusively present in cell nuclei. Protein expression proved to be extremely strong, with a H-score mean of 200.56. A strong signal was recorded in most or 45 samples (62.5%), while 19 samples (26.4%) had moderate and only eight samples had weak signal (11.1%). TWIST1 expression did not show statistically significant correlation with grade (*p* = 0.484), sex (*p* = 0.957), age (*p* = 0.249), or genetic changes of the *CDH1* gene (*p* = 0.772), *CDH2* gene (*p* = 0.919), or *CTNNB1* mutations (*p* = 0.146).

Immunohistochemical analysis of two proteins with similar transcriptional functions (SNAIL and SLUG), specified that their expression was most often recorded in the cytoplasm and in some samples in the nucleus. The expression of the protein signal in the cytoplasm was very strong, with a H-score mean of 190.00. The signal was recorded in all samples. Only eight samples (11.1%) showed a weak signal in the cytoplasm and all of them belonged to grade I. Likewise, all meningioma classified as grade III showed a strong signal. In most samples, the strongest signal was most often recorded, more precisely in 40 samples (55.6%). The expression of SNAIL and SLUG in nuclei was specifically quantified and classified into three categories: 1 = no signal in nuclei, 2 = signal in nuclei exists, and it is present in <50% of nuclei in the field of view and 3 = signal in nuclei is present in ≥50% of nuclei in the field of view (Figure 4). 

Many of the samples did not show protein expression in the nucleus (34 samples or 47.2%); however, samples in which more than 50% of immunopositive nuclei were recorded were atypical and anaplastic meningioma (six out of seven samples). This was supported by statistical analyses that showed that the increased expression of SNAIL and SLUG in the cytoplasm was associated with their increased expression in nuclei (χ^2^ = 17.795, df = 4, *p* = 0.001 and φc = 0.352; *p* = 0.001). There was a statistically significant difference in the expression of SNAIL and SLUG in different grades (F = 7.380, *p* = 0.001), which was confirmed by the Kruskal–Wallis test (H = 12.310, df = 2, *p* = 0.002). *Post-hoc* analysis found that there was a significant difference in the H-score means of grade I and grade II and grade I and grade III, indicating that the increase in grade was accompanied by increased protein expression. This suggests that protein expression in nuclei also increases with grade, which was confirmed by statistical tests (χ^2^ = 17.621, df = 4, *p* = 0.001, and φc = 0.350; *p* = 0.001). The cytoplasmic expression of SNAIL and SLUG proteins did not show statistically significant association with sex (*p* = 0.510), age (*p* = 0.399), or the studied genetic changes (*CDH1 p* = 0.781; *CDH2 p* = 0.503). Another significant association was observed between the expression of SNAIL and SLUG in the nucleus and the occurrence of genetic changes in the *CDH1* gene that were recorded by the microsatellite marker D16S3025 (χ^2^ = 16.573, df = 2, *p* = 0.000 and φc = 0.505; *p* = 0.000). That is, samples without *CDH1* mutations were significantly more likely to lack SNAIL and SLUG expression in the nucleus or it was expressed in less than 50% of the nuclei, while samples with mutations in the *CDH1* gene were significantly more likely to show protein expression in more than 50% of the nuclei. The expression of SNAIL and SLUG in nuclei also showed a statistically significant correlation with the occurrence of genetic changes in the pooled analysis of both microsatellite markers for both genes, the *CDH1* (*p* = 0.001) and the *CDH2* (D18S66 *p* = 0.002 or D18S819 *p* = 0.005). However, a more detailed *post-hoc* analysis could not confirm the statistical correlations of SNAIL and SLUG expression in nuclei with genetic changes being observed by both markers for the *CDH1* gene or with genetic changes of the *CDH2* gene recorded with the D18S66 or D18S819 marker. The association of SNAIL and SLUG expression with the presence of *CTNNB1* gene mutations was not statistically significant (*p* = 0.143).

### 3.5. Expression of Total and Non-Phosphorylated β-Catenin

The total β-catenin that was detecting both of its forms—phosphorylated and non-phosphorylated—was present at varying strengths in all samples of intracranial meningioma. Its expression was bound to the cytoplasm and membranes and it was not recorded in the nuclei. The mean value of the results that were obtained by the H-score was 91.50. A weak signal (58.3%) was mostly observed, while a moderate signal was recorded in 30.6% and strong signal in 11.1% samples. A statistically significant difference in protein expression of total β-catenin across different grades was established (F = 7.289, *p* = 0.001). This significance was also confirmed by the non-parametric Kruskal-Wallis test (H = 12.248, df = 2, *p* = 0.002). *Post-hoc* analysis showed that grades II and III had almost two times higher mean H-score value in respective to grade I, but higher grades did not differ from each other in the expression of total β-catenin. A significant correlation between protein expression and sex and age could not be established (*p* = 0.634 and *p* = 0.492). Furthermore, the protein expression of total β-catenin showed no statistical correlation with genetic changes in the *CDH1* gene (*p* = 0.965), *CDH2* gene (*p* = 0.780), or mutations in the *CTNNB1* gene (*p* = 0.545).

The protein expression of the non-phosphorylated or active form of β-catenin was observed in most samples of intracranial meningioma and it was also associated with the cytoplasm and membranes. The mean value of the H-score for active β-catenin expression was 85.43. The largest number of samples had low expression (61.1%), including three samples (4.6%) that lack signal. A strong signal was also recorded in low frequency, in six samples (8.3%), while moderate signal was observed in 30.6% of samples. Again, a statistically significant difference in protein expression of active β-catenin in different grades was established (F = 5.932, *p* = 0.004) and confirmed by the Kruskal–Wallis test (H = 12.021, df = 2, *p* = 0.002). In this case, as in the case of total β-catenin expression, grades II and III had almost two times higher mean H-score with respect to grade I, while higher grades did not differ from each other in the expression of non-phosphorylated β-catenin. The expression of non-phosphorylated β-catenin showed no statistically significant correlation with sex (*p* = 0.907), age (*p* = 0.419), or genetic changes in *CDH1* (*p* = 0.102), *CDH2* (*p* = 0.648), or *CTNNB1* (*p* = 0.422).

### 3.6. Comparison of E-, N-Cadherin, β-Catenin, TWIST1, SNAIL and SLUG Expression

The protein expressions were compared to gain insight into their interrelationships. Nine statistically significant correlations were found by comparing the protein expressions of E-cadherin, N-cadherin, both forms of β-catenin, TWIST1, SNAIL, and SLUG (Figure 5, Table 2). Protein expressions that proved to be statistically significantly correlated included the following pairs: E-cadherin and β-catenin (r = 0.373, *p* = 0.001), E-cadherin and the non-phosphorylated form of β-catenin (r = 0.247, *p* = 0.037), N-cadherin and β-catenin (r = 0.305, *p* = 0.009), N-cadherin and non-phosphorylated form of β-catenin (r = 0.277, *p* = 0.018), β-catenin and non-phosphorylated form of β-catenin (r = 0.628, *p* = 0.000), β-catenin and TWIST1 (r = 0.252, *p* = 0.033), non-phosphorylated form of β-catenin and SNAIL and SLUG (r = 0.279, *p* = 0.017), non-phosphorylated form of β-catenin and TWIST1 (r = 0.233, *p* = 0.048), and TWIST1 and SNAIL and SLUG (r = 0.251, *p* = 0.033). In all pairs, the correlation was positive.

## 4. Discussion

### 4.1. The Role of E- and N-Cadherin in Intracranial Meningioma

LOH of the 16q22.1 region, where the *CDH1* gene is located, is already described in different types of tumors, such as adenocarcinoma, nonseminoma, gallbladder, breast, endometrial, ovarian, prostate, and various gastrointestinal cancers [21,22,23,24,25,26,27]. The mutation of this region may result in metastasis and/or play a role in progression. The study of the *CDH1* gene in various brain tumors showed that 14.3% of primary brain tumors had LOH and that this genetic change was mainly related to meningioma (31% of meningioma samples showed LOH). Meningiomas with LOH belonged to the benign group, suggesting caution when predicting prognosis of the disease, since benign subtypes may hide invasive potential [28]. Our group previously showed similar results [29], who found the LOH of *CDH1* gene in 37.5% of meningioma. Simon et al. [30] suggest that the LOH of this gene contributes to the development of malignancies in meningioma. This study showed a slightly lower frequency of LOH in meningioma (in 15.3%), but also a tendency of a slight LOH frequency increase with higher grades, suggesting that the LOH of *CDH1* gene plays a role in meningioma progression. Furthermore, MSI also plays an important role in tumorigenesis and has been described in many tumors, primarily colorectal, gastric, and endometrial cancers [31,32]. Pykett et al. [33] claim that, according to their findings, MSI is involved in the tumorigenesis of meningioma. Pećina-Šlaus et al. [34] showed MSI of the *CDH1* gene in 11% of meningioma. The meningioma samples that were investigated in this study showed MSI more often than LOH. Although MSI was most commonly present in malignant grade with an incidence in 50% of samples, it was also reported in 22.5% of benign meningiomas. This finding also suggests that MSI of *CDH1* gene plays a role in meningioma progression. MSI findings in the *CDH1* gene may indicate increased genome instability, which, in the early stages of tumorigenesis, accelerates the process of tumor cell evolution by accumulating mutations [35]. A possible explanation for the lower frequency of MSI in atypical meningioma (11.7% of samples) is that such genome instability in the early stages of tumorigenesis can cause cell proliferation, but also their death if the mutation rate exceeds a certain threshold [36].

Genetic changes in the *CDH2* gene have not been studied as extensively in tumors as those of the *CDH1* gene. However, studies of chromosomal instability in various tumors, such as colorectal, bladder cancer, or head and neck squamous cell carcinoma have shown deletions of chromosome 18q that contains the N-cadherin gene [37,38,39,40]. The LOH of chromosome 18q has been shown to be involved in tumors with aggressive behavior and reduced survival rates [41]. As early as 1994, Jen et al. [42] suggested that the LOH of 18q has prognostic value for patients with colorectal cancer. Sarli et al. [43] consider the LOH of chromosome 18q to be a genetic marker indicating recurrence and survival rate for patients with stage III colorectal cancer. Recent studies have shown that the loss of the 18q11.2-q12.1 region should be considered when determining chemotherapy in patients with metastatic colorectal cancer [44]. Our study showed that the LOH of the *CDH2* gene was a more common event when compared to the LOH of the *CDH1* gene. Likewise, 61.9% of the samples with LOH also contained MSI (13/21 samples). Although the LOH of the *CDH2* gene was not significantly associated with any specific grade of meningioma, the samples recorded a slight increase of LOH through the grades, where 4/6 of the grade III samples (66.7%) contained allele loss. This finding suggests that, although the LOH of *CDH2* gene may be present in benign meningioma, its increase in higher grades may have a role in the progression of intracranial meningioma and the development of an aggressive character. It can be speculated that benign meningioma in which genetic changes in the *CDH1* and *CDH2* genes have been observed hide the genetic potential for eventual progression. MSI of the *CDH2* gene, with an incidence in 58.6% of samples, was shown to be a more frequent event when compared to the LOH of the *CDH2* gene (in 30% of samples) or compared to the MSI of the *CDH1* gene (in 22.2% of samples). MSI of *CDH2* gene was not statistically significantly associated with a particular grade, but it occurred at relatively similar frequencies across all stages of malignancy contributing to genome instability already in the early stages of intracranial meningioma development. It can be assumed that this constant finding of MSI contributes to the instability of the meningioma genome and it is responsible for the increase in the accumulation of mutations, which is reflected in the later stages of the disease.

Our samples of intracranial meningioma showed that, if the *CDH1* gene has a genetic change, it is very likely that the *CDH2* gene will also be altered. The observed accompanied changes can be explained by the constant occurrence of genomic instability in the samples that were investigated in this study. This is not surprising, given the fact that our previous studies [29] showed that MMR system in meningioma is often deficient, resulting in genomic instability. Those studies showed that 38% of meningioma had MSI in one locus, 16% in two loci, and 13.3% in three loci usually affecting one of genes responsible for correct functioning of MMR. Identifying specific tumor regions with MSI can have important implications in cancer diagnosis as well as potential in predicting disease course. The results of genomic instability in the investigated genes of this study suggest an increased frequency of mutations in meningiomas. The MSI in the genes that are responsible for the epithelial–mesenchymal transition most likely has its roots in the instability of the genes of the MMR system. Genetic changes of *CDH1* and *CDH2* are present in all meningioma grades. Their correlation suggests the accumulation of genetic changes in the genes that are responsible for EMT and establishment of defective EMT in meningioma tumors, regardless of grade. Additionally, the *CDH1* and *CDH2* genes were often simultaneously changed in same samples. Therefore, we can speculate that *CDH1* and *CDH2* genes may represent a useful prognostic marker of meningioma progression, especially since genetic changes occur in benign stages and increase their frequency by transitioning to malignant forms (especially LOH in the *CDH2* gene).

The results of the protein-level analysis presented in this study contributes to further elucidation of the E- and N-cadherin role in intracranial meningioma. The expression of E-cadherin in intracranial meningioma showed weak expression with a H-score mean value of 48.83 and with 30.6% samples in which no signal was recorded. Although genetic changes in its *CDH1* gene did not show a statistically significant association with low expression levels, 9/11 samples with LOH (81.8%) and 13/16 samples with MSI (81.2%) had low expression, indicating that overall genetic changes contribute to the decreased expression. Several studies observed that some samples of prostate, bladder, endometrial, hereditary diffuse gastric cancer, and invasive ductal or lobular carcinoma did not contain a genetic change in *CDH1*, but showed a decrease in E-cadherin expression. In these cases, the loss of E-cadherin expression can be attributed to transcriptional or posttranslational modifications, or epigenetic attenuation, such as methylation of the E-cadherin promoter [26,45]. The protein levels of E-cadherin in our study were not statistically associated with grade, which indicated that weakened E-cadherin expression may be an early and constant event. Novel studies confirmed that partial EMT, where E-cadherin expression is not decreased, also has a role in invasiveness of cancer. Thanks to partial EMT, tumors more easily adapt to a new microenvironment and development of a secondary tumor, causing treatment ineffectiveness and relapse [11,12].

N-cadherin expression in the studied samples had a mean H-score expression value of 81.04. Although expression did not show a statistically significant association with any grade, it is evident that it was higher relative to E-cadherin expression levels. It is known from the literature that the increased expression of N-cadherin can be a trigger of the aggressive character of the tumor with the possibility of metastasis and higher recurrence rates. It has been reported in tumors, such as breast, lung, prostate, pancreatic, melanoma, multiple myeloma, gastric carcinoma, or oral squamous cell carcinoma [46,47]. In breast tumors, N-cadherin promotes cell motility independent of E-cadherin expression [47]. Moreover, N-cadherin expression may be a prognostic marker of glioma metastasis, as it may overcome the inhibitory effect of E-cadherin [47]. A statistically significant correlation was found between N-cadherin expression and genetic changes in the *CDH1* gene (E-cadherin) (*p* = 0.016). *Post-hoc* analysis showed that samples with the mutation on E-cadherin had, on average, significantly lower H-score expression of N-cadherin. Furthermore, 73.7% of samples with poor N-cadherin expression had a change in the *CDH2* gene, while samples with a strong signal had fewer genetic changes (in 33.3% of samples). We can speculate that the overall genetic changes in *CDH2* contribute to a decreased expression of its protein product. It would be interesting for future studies to investigate potential activating mutations of this gene.

Although the protein expressions of E- and N-cadherin did not show a statistically significant difference, comparing the H-score mean values of their expressions (48.83 versus 81.04), a higher expression of N-cadherin is notable. Our findings suggest a cadherin change (the so-called cadherin switch) in intracranial meningioma that is a prominent feature of EMT, independent of grade. A possible explanation for the high expression of N-cadherin in grade I is that even some benign meningioma subtypes may contain an invasive character with the potential for progression and a higher recurrence rate like higher grades. Because of all the above, N-cadherin could potentially be a good prognostic marker of meningioma behavior regardless of grading.

Several studies investigated other markers of mesenchymal phenotype and collectively showed an upregulation of those proteins in meningioma as compared to the normal levels. Immunohistochemical examination demonstrated that meningioma cells were positive for vimentin staining and they noted strong diffuse expression in 15 meningiomas they investigated [48]. Ng and Wong [49] inspected the use of immunohistochemistry Cryo-sections from 50 meningiomas of diverse histological subtypes and found vimentin positive in 98% of meningiomas. Furthermore, Sharma et al. [50] further strengthened those findings by using multiple quantitative proteomic and immunoassay-based approaches for the serum proteomic analysis of different grades of meningiomas, which found that vimentin exhibited a very high level of differential expression in the meningioma patients as compared to the healthy subjects and that the increase of vimentin was associated to higher malignancy grades. Other important mesenchymal markers were also found to be expressed in meningioma. Western blot and immunohistochemistry analysis found the expression of laminin γ1 protein. The study also showed that it was higher in grade III meningiomas than in grade I meningiomas. Additionally, higher levels of laminin γ1 were associated with a significantly shorter tumor recurrence time and decreased patient survival time [51]. Wallesch et al. [15] found the difference in the expression levels of transcription factor Zeb-1, which suppresses E-cadherin, according to the malignancy grade of meningiomas. Immunohistochemistry and real-Time-PCR, performed on 85 cases of various histopathological subtypes and grades of malignancy found significantly increased expression of Zeb-1 associated to aggressive WHO grade II or III meningiomas. This was accompanied with the E-cadherin downregulation in high grade meningiomas. Furthermore, the study indicated that reduced E-cadherin levels were more pronounced in recurrent as compared to non-recurrent meningiomas. Another investigation utilizing microarray analysis and bioinformatics also revealed that ZEB1 was significantly upregulated in malignant meningioma tissues and that its regulation involved miR-4652-3p [52].

### 4.2. The Role of SNAIL, SLUG and TWIST1 Transcription Factors in Intracranial Meningioma

It is known from the literature that enhanced expressions of SNAIL and TWIST1 proteins are reliable markers of aggressive character or invasiveness of various tumors, such as breast cancer, oropharyngeal squamous cell carcinoma, colorectal cancer, and odontogenic tumors [53,54,55]. The increased expression of transcription factors is often an indicator of a poorer disease outcome or resistance to therapy and is, therefore, considered to be a prognostic marker of hematologic tumors, invasive lobular carcinoma, mandibular squamous cell carcinoma, and colon carcinoma [56,57,58,59]. Given the opposite results of many published papers on the role of the expression of EMT transcription factors in various tumors, Zhang et al. [60] decided to make a meta-analysis of all data. The results of this meta-analysis of the expression of SNAIL and TWIST1 transcription factors in different tumors confirmed that their increased expression affects tumor development and suggests an unfavorable treatment outcome, with TWIST1 being a better prognostic marker than SNAIL. The results of the present study proved SNAIL and SLUG expression to be better prognostic markers of meningioma progression as compared to TWIST1, even though their expressions were statistically significantly positively associated (*p* = 0.033). The finding that SNAIL and SLUG are better prognostic markers of tumor progression compared to TWIST1 is also consistent with the results of Wallesch et al. [15] It is known that higher grades of astrocytic tumors (especially glioblastomas) are associated with a stronger expression of the TWIST1 protein, which is an independent predictor of poorer treatment outcomes [61]. Nagaishi et al. [62] noted an increased expression of the transcription factors SLUG and TWIST1 with decreased regulation of E-cadherin in meningeal tumors of solitary fibrous tumors and hemangiopericytomas. Wallesch et al. [15], by comparing protein expression in higher-grade meningioma with lower meningioma, recorded a decrease in E-cadherin expression accompanied by an increase in SLUG protein expression, but also a slight decrease in TWIST1 protein expression. The protein expression of TWIST1 in this study proved to be extremely strong with a H-score mean of 200.56. No anaplastic meningioma showed a weak signal and 66.7% showed a very strong signal.

The results of SNAIL and SLUG protein expression also showed high expression levels with a H-score mean of 190.00 in the cytoplasm. All the samples (11.1%) that showed a weak signal in the cytoplasm were classified as grade I. However, all the meningioma classified as grade III showed a strong signal. Except in the cytoplasm, in part of the samples the expression of SNAIL and SLUG proteins was in the nuclei (52.8%). Statistical analysis showed that the increased expression of SNAIL and SLUG in the cytoplasm leads to an increase in their expression in nuclei (*p* = 0.001). Further analysis showed a statistically significant association of SNAIL and SLUG protein expression with different grades (*p* = 0.001), which was also true for protein expression in nuclei (*p* = 0.001), suggesting its accumulation and role in higher grades. This result is consistent with other studies that also report that the expression of SNAIL and SLUG in nuclei was associated with higher malignant potential and poorer prognosis in other tumors, such as malignant pleural mesothelioma and in lung neuroendocrine tumors [63,64]. Despite the high expression of the transcription factors TWIST1, SNAIL, and SLUG relative to the expression of E-cadherin, no significant negative correlation was observed, which is partly in line with the findings of the research group of Wallesch et al. [15]. However, we showed extremely strong expressions of SNAIL, SLUG and TWIST1 in relation to the low expression of E-cadherin so their regulatory role in meningioma cannot be ruled out. Additionally, as already mentioned, this study showed that *CDH1* genetic changes may contribute to decreased E-cadherin expression. The expression of SNAIL and SLUG proteins in the nucleus was associated with the occurrence of genetic changes in the gene *CDH1* recorded by the microsatellite marker D16S3025 (*p* = 0.000). It is possible that, in addition to genetic changes, transcription factors SNAIL, SLUG, and TWIST1, which repress E-cadherin synthesis by binding to its promoter, further contribute to the reduction in E-cadherin expression levels in intracranial meningioma samples.

It is known that the enhanced expression of EMT transcription factors does not necessarily represent a late event in various cancers, but it may also be involved in the initial events of tumor development [65]. The results of this study are consistent with this claim, since we also observed strong expression of SNAIL, SLUG, and TWIST1 in some cases of benign meningioma, reaffirming that meningioma, although histologically classified as benign, may hide an invasive character due to their molecular landscape, activated pathways, and driving mutations, which can lead to their progression. Thanks to all the above, SNAIL, SLUG, and TWIST1 could contribute to better patient stratification in the treatment of intracranial meningioma. The expression of SNAIL and SLUG in nuclei can be used as an additional prognostic marker for the progression of meningioma.

### 4.3. Role of β-Catenin Phosphorylation Status and Wnt Signaling Pathway in Intracranial Meningioma

This study showed that there is a relatively high frequency of mutations in exon 3 of β-catenin. Exon 3 has been known as a mutational hot spot in many tumor types, including breast, colon, liver, prostate, and melanoma, contributing toβ-catenin oncogenic potential [66]. Our findings correspond to those of Kim and Jeong [67], who used the available Human Cancer Genome Databases. When considering different carcinomas, mutations of exon 3 occurred in the frequency of 0–36% of the samples. Mutations were most common in endometrial and liver cancers, while they were the least common in colorectal cancers. Mutations in the *CTNNB1* gene of endometrial cancers are more common in the lower grades and early stages of disease development, and they are associated with a poorer disease outcome due to a higher chance of recurrence. Because our study did not record mutations in the highest, malignant grade of meningioma, but mainly in benign grades, it is possible that these mutations represent the driving mutations of meningioma and they are not related to progression. Furthermore, Lee et al. [68] studied the most common mutations of exon 3 at the phosphorylation sites S33, S37, S41, and T45 in various brain tumors, including meningioma. They showed that the codons for these critical amino acids were not altered in any primary tumor, including meningioma, but the mutation at S33 was only found in the metastasis of the primary lung cancer tumor. The results of our study showed a similar scenario where these codons did not harbor mutations. Mutations were found in other codons within the same exon, which may indicate nucleotide changes specific to meningioma. Immunohistochemical analysis of mutated samples was performed using two antibodies that monitor total β-catenin (phosphorylated and non-phosphorylated) and the non-phosphorylated or active form of β-catenin (NON-P β-catenin) showed. The NON-P antibody does not detect β-catenin phosphorylated at S33/S37/T41, which was also showed in the study by Sakai et al. [69]. Although in some samples with mutations, which harbor a change in amino acid sequence or premature cessation of protein synthesis resulting in the protein lacking phosphorylation sites for kinases in its N-terminal domain, which is recognized by destruction complex [66], the active form of β-catenin was weak (83.3%) or moderate in signal (16.7%). However, in most samples with mutations (78.5%) the active form showed the same signal relative to total β-catenin, which suggests that cases with mutations on *CTNNB1* are mainly producing active form that leads to the activation of the Wnt signaling pathway. It should also be noted, that in this study, mutated samples of atypical meningiomas showed a moderate signal of both forms of β-catenin in 75% of cases.

Immunohistochemical analysis for total β-catenin and NON-P β-catenin on all meningioma samples showed very close mean values of their H-score (91.50 for total β-catenin and 85.43 for NON-P β-catenin), leading to a significant association between the expression of β-catenin and its non-phosphorylated form (*p* = 0.000) and confirming the presence of the active mutant form of the protein. Additionally, this study proved that the increase in the expression of total β-catenin is accompanied by an increase in the expression of the active form, and that the increase in the expression of both forms of β-catenin is statistically significantly associated with higher degrees. The literature states that increased expression of β-catenin may be due to malfunction of the destruction complex, increased expression of WNT ligands as well as loss or reduction of regulation [70]. The results of active β-catenin expression in the grade II and grade III samples of this study suggest that the Wnt signaling pathway plays an important role in meningioma and their progression. However, the absence of β-catenin expression in the nuclei of the samples suggests that, although β-catenin is active, there is the possibility that, in benign cases, it has not yet reached the nucleus. The absence of β-catenin expression in higher grade malignant meningioma nuclei is difficult to explain, but it can be explained by the small number of samples in these groups. This is supported by the results of Shimada et al. [71] and Brunner et al. [72], who did not record the expression of β-catenin in the cell nucleus of meningioma, and the results of Rutkowski et al. [73], who did not record nuclear expression of β-catenin in all higher-grade samples. In the work of Rutkowski et al. [73] antibody that captures the C-terminal end of β-catenin was used, and nuclear expression was recorded in 33/50 higher grade tumors with only 6/50 tumors showing strong expression in almost all cells.

Given the high expression of the active form of β-catenin at higher grades, one can speak of the role of the Wnt signaling pathway in the progression of intracranial meningioma. To finally confirm this hypothesis, research is needed on a bigger number of higher-grade meningioma samples, with the aim of elucidating the transition of β-catenin to the nuclei.

### 4.4. Interaction of Wnt Signaling Pathway and EMT in Intracranial Meningioma

It is known that the translocation of β-catenin into the nucleus can lead to the decreased expression of E-cadherin and induce tumor invasiveness. However, in our samples, the samples with higher expression of E-cadherin were associated with the higher expression of total and active β-catenin (*p* = 0.001 and *p* = 0.037). Although many studies speak of E-cadherin as a tumor suppressor, recent research emphasizes that E-cadherin expression maintains the phenotype of tumor cells, even those with an invasive character. E-cadherin expression does not have to fall significantly for EMT to occur [74,75]. Furthermore, in our study, the expression of E-cadherin and β-catenin is not as pronounced in membranes as in the cytoplasm, which suggests the breakdown of the membrane E-cadherin/β-catenin complex that is necessary for cell adhesion. Likewise, the total expression of E-cadherin was weaker as compared to the expression of β-catenin, which was mainly non-phosphorylated.

The total expression of N-cadherin was somewhat closer and significantly associated with the expression of β-catenin (*p* = 0.009, *p* = 0.018), which fits the claim that the active form of β-catenin contributes to the expression of N-cadherin and EMT. It is also known that the N-cadherin/β-catenin complex is a source of β-catenin in cancer cells required for the transcription of the oncogene in the nucleus [46].

The results of the increased expression of each transcription factor, TWIST1, SNAIL, and SLUG, accompanied by increased β-catenin levels support the fact that β-catenin accumulation stimulates the transcription of SNAIL, SLUG, and TWIST1, as well as the fact that the Wnt signaling pathway is most likely activated by impossibility of GSK3β to phosphorylate SNAI1. Thus, β-catenin is stabilized, and E-cadherin repression is promoted. This creates a closed cycle, because the repression of E-cadherin allows for the accumulation of β-catenin in the cytoplasm and nucleus, thus encouraging the transcription of various oncogenes [76,77]. Therefore, the results of this investigation support a close association between Wnt signaling pathway activation and epithelial-mesenchymal transition in intracranial meningioma.

## 5. Conclusions

This study sheds light and brings novel insights on the role of important molecules of the Wnt signaling pathway and EMT in the development and progression of intracranial meningioma. We showed that genes involved in cadherin switch, *CDH1* and *CDH2,* may play a role in the development and progression of meningiomas, where both genes are involved and the basis for defective EMT has been established. Protein analyses also revealed weaker expression of E-cadherin in relationship to N-cadherin, suggesting EMT in meningioma. Our investigation on EMT transcriptional factors TWIST1, SNAIL, and SLUG demonstrated their strong expression, where SNAIL and SLUG were associated to higher grades and they represent a good indicator of the progression of intracranial meningiomas. The results on β-catenin showed that it is mostly present in its active form. Additionally, the active form of β-catenin was significantly associated to higher grades, suggesting the role of the Wnt signaling pathway as a driver of meningioma progression. Our protein analyses of E-cadherin, N-cadherin, SNAIL, SLUG, and TWIST1 and two forms of β-catenin demonstrated a close association between the Wnt signaling pathway activation and epithelial-mesenchymal transition in intracranial meningiomas.

Recognizing the molecular changes that are responsible for cell motility control brings potential markers of progression, discovers new molecular targets for therapeutic interventions in malignant forms of intracranial meningioma, and helps to accurately diagnose meningioma by selecting those patients who may show an unfavorable course. The detection of the role of the Wnt signaling pathway and EMT in the development of more aggressive histopathological types of meningioma ultimately contributes to increased survival and better prospects of patients with intracranial meningioma.

## Figures and Tables

**Figure 1 cancers-13-01633-f001:**
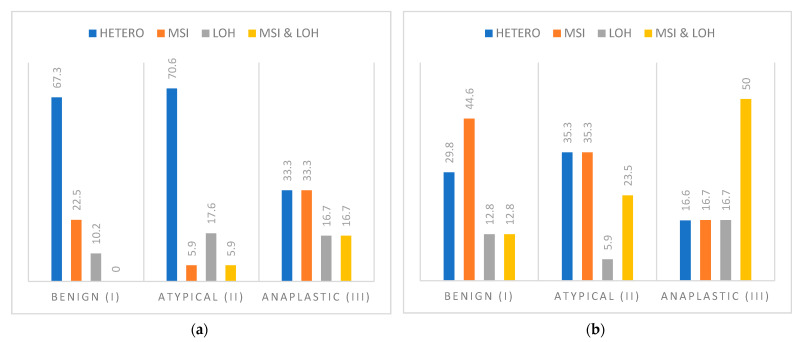
(**a**) The results (in percentages) of the pooled analysis of two microsatellite markers D16S752 and D16S3025 for the *CDH1* gene in different grades of meningioma. Legend: HETERO—a heterozygote that has no genetic change; MSI—microsatellite instability; LOH—loss of heterozygosity. (**b**) Results (in percentages) of the pooled analysis of two microsatellite markers D16S752 and D16S3025 for the *CDH2* gene in different grades of meningioma. Legend: HETERO—a heterozygote that has no genetic change; MSI—microsatellite instability; LOH—loss of heterozygosity.

**Figure 2 cancers-13-01633-f002:**
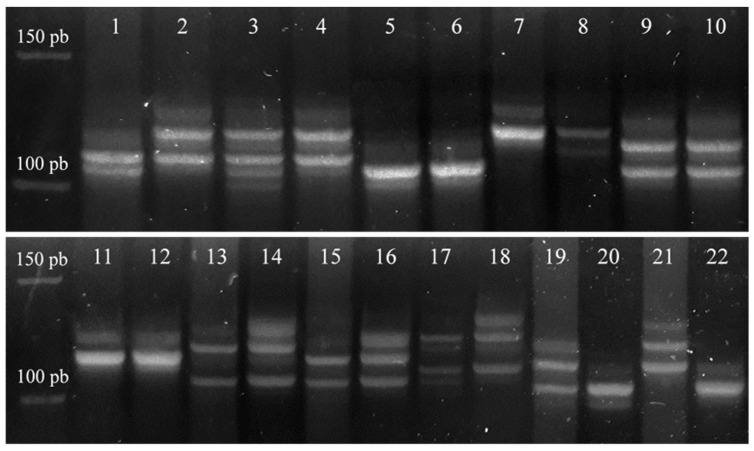
Examples of genetic changes in the *CDH1* and *CDH2* genes recorded by the microsatellite markers in meningioma samples on Spreadex gels (Elchrom Scientific, AL-Diagnostic GmbH, Amstetten, Austria). Odd numbers represent tumor DNA while even numbers represent the blood DNA of the same patient. Stripes 1 and 2—the sample shows MSI; lanes 3 and 4-the sample shows MSI; stripes 5 and 6—heterozygous sample without changes; lanes 7 and 8—the sample shows MSI; stripes 9 and 10—heterozygous sample without changes; stripes 11 and 12—heterozygous sample without change; stripes 13 and 14—the sample shows LOH; lanes 15 and 16—the sample shows LOH; stripes 17 and 18—the sample shows LOH; lanes 19 and 20—the sample shows MSI; lanes 21 and 22—the sample shows MSI.

**Figure 3 cancers-13-01633-f003:**
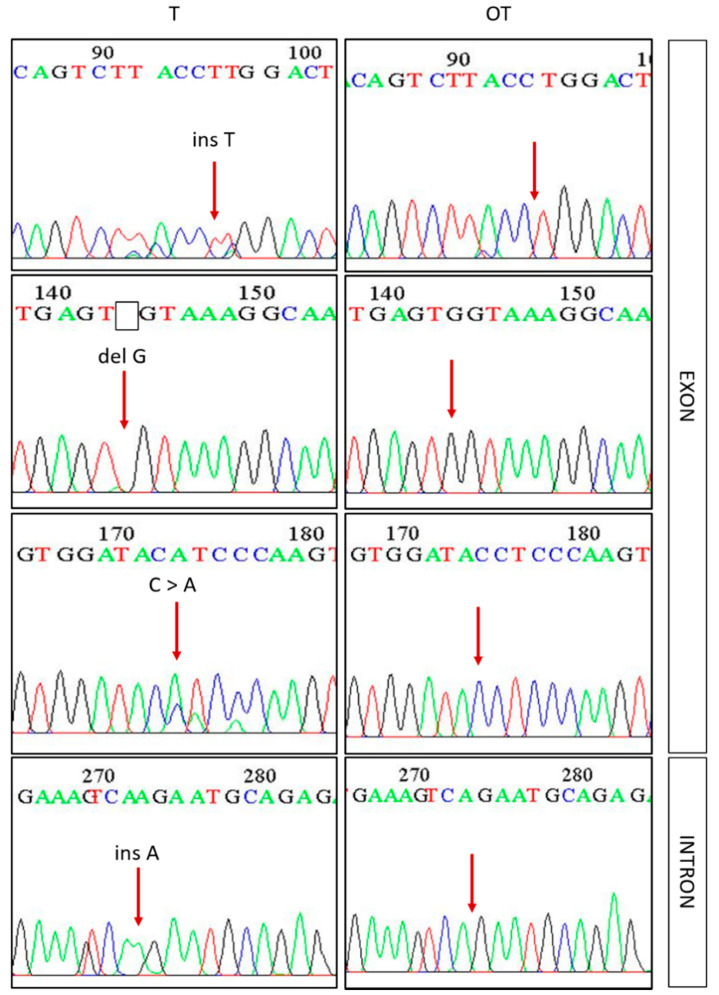
Examples of sequences with different types of mutations in exon 3 and intron 3 of the *CTNNB1* gene obtained by the Sanger sequencing method. A red arrow indicates the location of the mutation. Legend: T—tumor sample, OT—blood sample of the same patient.

**Figure 4 cancers-13-01633-f004:**
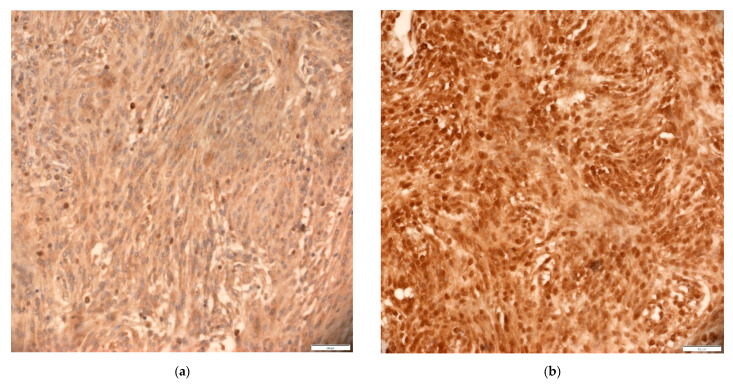
Immunohistochemical staining of meningiomas with polyclonal SNAIL and SLUG antibody at 200× magnification. (**a**) sample with low nuclei staining intensity (<50% of nuclei) and (**b**) sample with strong nuclei staining intensity (≥50% of nuclei). Scale bars: 50 μm.

**Figure 5 cancers-13-01633-f005:**
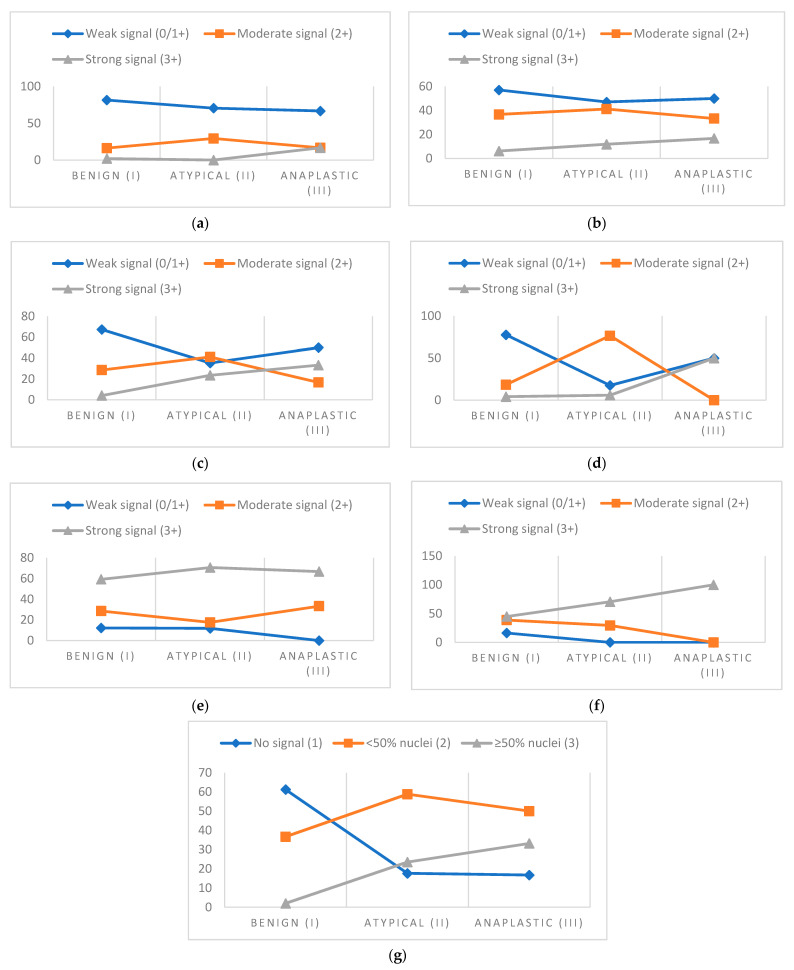
Percent of signal strength in different grades of meningioma shown for (**a**) E-cadherin, (**b**) N-cadherin, (**c**) total β-catenin, (**d**) NON-P β-catenin, (**e**) TWIST1, (**f**) SNAIL & SLUG in cytoplasm, and (**g**) SNAIL & SLUG in nuclei.

**Table 1 cancers-13-01633-t001:** Polymerase chain reaction conditions for amplification of *CDH1* and *CDH2* microsatellite markers and exon 3 of the *CTNNB1* gene.

Gene	(Pre)Denaturation	Denaturation	Annealing	Extending	No. of Cycles
D16S752 (*CDH1*)	96 °C/3 min	96 °C/30 s	55 °C/35 s	72 °C/30 + 1 s	35
D16S3025 (*CDH1*)	95 °C/10 min	95 °C/45 s	48 °C/30 s	72 °C/1 min	30
D18S66 (*CDH2*)	95 °C/5 min	95 °C/35 s	56 °C/35 s	72 °C/30 + 1 s	35
D18S819 (*CDH2*)	95 °C/5 min	95 °C/35 s	56 °C/35 s	72 °C/30 + 1 s	35
*CTNNB1* (exon 3)	96 °C/3 min	96 °C/30 s	55 °C/35 s	72 °C/30 + 1 s	40

**Table 2 cancers-13-01633-t002:** Results of correlations between protein expressions of E-cadherin, N-cadherin, β-catenin, non-phosphorylated β-catenin (NON-P β-catenin), TWIST1 and SNAIL and SLUG. Pairs that showed a statistically significant association were: E-cadherin and β-catenin (*p* = 0.001), E-cadherin and NON-P β-catenin (*p* = 0.037), N-cadherin and β-catenin (*p* = 0.009), N-cadherin and NON-P β-catenin (*p* = 0.018), β-catenin and NON-P β-catenin (*p* = 0.000), β-catenin and TWIST1 (*p* = 0.033), NON-P β-catenin and SNAIL and SLUG (*p* = 0.017), NON-P β-catenin and TWIST1 (*p* = 0.048), and TWIST1 and SNAIL and SLUG (*p* = 0.033). In all pairs, the correlation was positive.

Correlations		E-Cadherin	N-Cadherin	NON-P β-Catenin	β-Catenin	SNAIL & SLUG	TWIST1
E-cadherin	Pearson Correlation	1	0.140	0.247 *	0.373 **	0.227	0.005
	Sig. (2-tailed)		0.240	**0.037**	**0.001**	0.056	0.970
	N	72	72	72	72	72	72
N-cadherin	Pearson Correlation	0.140	1	0.277 *	0.305 **	−0.008	−0.035
	Sig. (2-tailed)	0.240		**0.018**	**0.009**	0.948	0.769
	N	72	72	72	72	72	72
NON-P β-catenin	Pearson Correlation	0.247 *	0.277 *	1	0.628 **	0.279 *	0.233 *
	Sig. (2-tailed)	**0.037**	**0.018**		**0.000**	**0.017**	**0.048**
	N	72	72	72	72	72	72
β-catenin	Pearson Correlation	0.373 **	0.305 **	0.628 **	1	0.204	0.252 *
	Sig. (2-tailed)	**0.001**	**0.009**	**0.000**		0.086	**0.033**
	N	72	72	72	72	72	72
SNAIL & SLUG	Pearson Correlation	0.227	−0.008	0.279 *	0.204	1	0.251 *
	Sig. (2-tailed)	0.056	0.948	**0.017**	0.086		**0.033**
	N	72	72	72	72	72	72
TWIST1	Pearson Correlation	0.005	−0.035	0.233 *	0.252 *	0.251 *	1
	Sig. (2-tailed)	0.970	0.769	**0.048**	**0.033**	**0.033**	
	N	72	72	72	72	72	72

*. Correlation is significant at the 0.05 level (2-tailed). **. Correlation is significant at the 0.01 level (2-tailed). Bold numbers indicate significant correlation.

## Data Availability

Data supporting reported results are contained within the article. Some of the data presented in this study are available on request from the corresponding author. The data are not publicly available due to privacy issues.

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
