# Peer review of "Are We Benign? What Can Wnt Signaling Pathway and Epithelial to Mesenchymal Transition Tell Us about Intracranial Meningioma Progression"

_cancers, 2021, doi:10.3390/cancers13071633_

Round 1
Reviewer 1 Report
The manuscript entitled “Are we benign? What can Wnt signaling pathway and epithelial to mesenchymal transition tell us about intracranial meningioma progression” demonstrated the relationship between WNT and EMT pathways and their role in intracranial meningioma. The reviewer has a few comments which the authors may address:
- The authors should increase the sample size to increase the precision of their estimates, which means that, for any given estimate/size of effect, the greater the sample size the more “statistically significant” the result will be.
- The authors are suggested to analyze the two microsatellite markers D18S66 and D18S819 for the CDH2 gene in different grades of meningioma.
- The authors are suggested to discuss the correlation between CDH1 and CDH2 in genetic changes.
- What is the correlation between protein expression of Laminin, Vimentin, ZEB1 with grade, sex, age, or genetic changes of CDH1, CDH2, or CTNNB1 mutations?
- What is the relationship between mutations in exon 3 of β-catenin and β-catenin phosphorylation and Wnt signaling pathway in progression?
- How did the authors measure H-score?
- The authors are suggested to investigate the underlying mechanisms for statistically significantly correlated proteins. Bioinformatic analysis is not sufficient.
- Some grammatical errors and typos are present in the manuscript that needs correction. The authors are advised to proofread the paper by a native English speaker and modify the errors.
Author Response
Comments of Reviewer #1:
Rviewer 1 commented that the manuscript demonstrated the relationship between WNT and EMT pathways and their role in intracranial meningioma. The reviewer had several comments which we addressed:
- The authors should increase the sample size to increase the precision of their estimates, which means that, for any given estimate/size of effect, the greater the sample size the more “statistically significant” the result will be.
Answer: Sample size was determined based on tumor incidence, financial considerations, and on similarity to other studies of similar size in the investigated field. There are many published studies on similar sample sizes and only big project consortia such as TCGA Research Network atlas can afford the large-scale microarray use for large number of samples. Nevertheless, the information we bring is very detailed and our findings were consistent. We would like to point out that in terms of our country, in which the study was conducted, the sample size of 72 meningiomas is quite a challenge to collect. Although meningiomas are frequent primary brain tumors, the occurrence of grades II and III is very low (20% and 1-3% of all meningioma in population). We have collected samples for period of several years from three major national hospitals and still couldn’t collect more because of the low incidence in our small population of 4 million people. Also because of novel techniques in surgery, some samples were not available or suitable for genetic and protein analysis. For the types of molecular studies that we conducted we believe that the number of investigated patients is adequate.
- The authors are suggested to analyze the two microsatellite markers D18S66 and D18S819 for the CDH2 gene in different grades of meningioma.
Answer: We have analyzed the two markers for CDH2 gene. However, we presented the pooled results for both markers. Frequencies for each single marker in different grades did not show statistical associations for specific grade or type. This was indicated on page 6, lines 245-248 where we wrote and also added an explanatory sentence:” The pooled analysis of CDH2 genetic changes also did not show a statistically significant association of the frequency of these changes with sex (p = 0.620), age (p = 0.151), or grade (p = 0.307). However, it is notable from the Figure 1 that anaplastic cases harbored the highest percent of MSI and LOH.” CDH2 gene changes were noted in more than 70% of samples across all grades thus statistical association to any meningioma grade was not established.
- The authors are suggested to discuss the correlation between CDH1 and CDH2 in genetic changes.
Answer: In discussion section on page 16, lines 542 there is a paragraph concentrated on correlation between genetic changes of CDH1 and CDH2 gene: “Our samples of intracranial meningioma showed that if the CDH1 gene has a genetic change, it is very likely that the CDH2 gene will also be altered. …etc.” We think those accumulated genetic changes are important and we reorganized the paragraph in order to more thoroughly discuss them in terms of correlation. The new paragraph is as follows: “Our samples of intracranial meningioma showed that if the CDH1 gene has a genetic change, it is very likely that the CDH2 gene will also be altered. The observed accompanied changes can be explained by the constant occurrence of genomic instability in the samples investigated in this study. This is not surprising, given the fact that our previous studies [29] showed that MMR (mismatch repair) system is often deficient in meningioma, resulting in genomic instability. Those studies showed that 38% of meningioma had MSI in one locus, 16% in two loci and 13.3% in three loci usually affecting one of genes responsible for correct functioning of MMR. Identifying specific tumor regions with MSI can have important implications in cancer diagnosis as well as potential in predicting disease course. The results of genomic instability in the investigated genes of this study suggest an increased frequency of mutations in meningiomas. The MSI in the genes responsible for the epithelial-mesenchymal transition most likely has its roots in the instability of the genes of the MMR system. Genetic changes of CDH1 and CDH2 were present in all meningioma grades. Their correlation suggests the accumulation of genetic changes in the genes responsible for EMT and establishment of defective EMT in meningioma tumors regardless of grade. Also, the CDH1 and CDH2 genes were often simultaneously changed in same samples. Therefore, we can speculate that CDH1 and CDH2 genes may represent a useful prognostic marker of meningioma progression especially since genetic changes occur in benign stages and increase their frequency by transitioning to malignant forms (especially LOH in the CDH2 gene).”
- What is the correlation between protein expression of Laminin, Vimentin, ZEB1 with grade, sex, age, or genetic changes of CDH1, CDH2, or CTNNB1 mutations?
Answer: We were not investigating those particular mesenchymal markers, however this is an excellent point raised by the reviewer and we decided to inspect the results of other studies investigating Laminin, Vimentin and ZEB1. The following paragraph was added in the Discussion section on page 18, line 606: "Several studies investigated other markers of mesenchymal phenotype and collectively showed upregulation of those proteins in meningioma as compared to the normal levels. Immunohistochemical examination demonstrated that meningioma cells were positive for vimentin staining and noted strong diffuse expression in 15 meningiomas they investigated (Zhao et al, 2015). (https://www.ncbi.nlm.nih.gov/pmc/articles/PMC4443052/
Ng and Wong (1993) inspected using immunohistochemistry Cryo-sections from 50 meningiomas of diverse histological subtypes and found vimentin positive in 98% of meningiomas. https://onlinelibrary.wiley.com/doi/abs/10.1111/j.1365-2559.1993.tb00089.x?sid=nlm%3Apubmed Furthermore, Sharma et al, (2014) https://pubmed.ncbi.nlm.nih.gov/25413266/ further strenghtened those findings by using multiple quantitative proteomic and immunoassay-based approaches for the serum proteomic analysis of different grades of meningiomas which found that vimentin exhibited very high level of differential expression in the meningioma patients as compared to the healthy subjects and that the increase of vimentin was associated to higher malignancy grades. Other important mesencymal markers were also found to be expressed in meningioma. Western blot and immunohistochemistry analysis found the expression of laminin γ1 protein. The study also showed that it was higher in grade III meningiomas than in grade I meningiomas. Also, higher levels of laminin γ1 were associated with a significantly shorter tumor recurrence time and a decreased patient survival time (Ke et al, 2013). https://pubmed.ncbi.nlm.nih.gov/23053286/
Wallesch et al, (2017) https://pubmed.ncbi.nlm.nih.gov/28870549/ found the difference in the expression levels of transcription factor Zeb-1, that suppresses E-cadherin, according to the malignancy grade of meningiomas. Immunohistochemistry and real-Time-PCR, performed on 85 cases of various histopathological subtypes and grades of malignancy found significantly increased expression of Zeb-1 associated to aggressive WHO grade II or III meningiomas. This was accompanied with the E-cadherin downregulation in high grade meningiomas. Furthermore, the study indicated that reduced E-cadherin levels were more pronounced in recurrent as compared to non-recurrent meningiomas. Another investigation using microarray analysis and bioinformatics also revealed that ZEB1 was significantly upregulated in malignant meningioma tissues and that its regulation involved miR-4652-3p (Li et al, 2019). https://reader.elsevier.com/reader/sd/pii/S0753332219303786?token=3DEA669729DBEBEA55CC376FB20113D9F4C9B504F10C9F3659EA6FCEC85A9E9008642248F76878AFD0757E1E71A1C75C
- What is the relationship between mutations in exon 3 of β-catenin and β-catenin phosphorylation and Wnt signaling pathway in progression?
Answer: The relationship between mutations in exon 3 of β-catenin and β-catenin phosphorylation was statistically analyzed and showed no significant correlation. However, we investigated samples with mutation and noticed that in most samples (13 of 14) the signal of the non-phosphorylated (active) form of β-catenin was equal to the signal of total β-catenin suggesting that mutated samples harbored mostly active form of β-catenin. We agree that this was not stated clearly in discussion, so we are adding an explanation to the article:
“Although in some samples with mutations, which harbor a change in amino acid sequence or premature cessation of protein synthesis resulting in the protein lacking phosphorylation sites for kinases in its N-terminal domain which is recognized by destruction complex [66], the active form of β-catenin was weak (83.3%) or moderate in signal (16.7%). However, in most samples with mutations (78,5%) the active form showed the same signal relative to total β-catenin, suggesting that cases with mutations on CTNNB1 are mainly producing active form that leads to the activation of the Wnt signaling pathway. It should also be noted, that in this study, mutated samples of atypical meningiomas showed a moderate signal of both forms of β-catenin in 75% of cases.”
- How did the authors measure H-score?
Answer: In material and methods section 2.3. Protein localization and expression on page 5, lines 189-, we described how we measured H-score:
“To evaluate immunopositivity of each sample, minimal number of 200 cells were counted in the tumor hotspot. Cells with specific expression level were counted using ImageJ software (National Institutes of Health, USA) and evaluated by a semiquantitative method. The quantification was done with H-score (Histo-score) which includes the sum of individual H-scores for each intensity and gives a higher value to higher intensities of protein expression:
H = [1 × (% cells 1+) + 2 × (% cells 2+) + 3 × (% cells 3+)]
where 1+ indicates weak immunopositivity - yellowish / light brown color, 2+ indicates moderate immunopositivity - light brown and 3+ indicates strong immunopositivity - dark brown. By calculating the H-score, a range of protein expression values on a scale of 0–300 was obtained [20]. Based on the H-score value, protein expression of the samples was categorized into three categories of signal strength: 0-100 = no signal/weak signal (0/1+), 101-200 = moderate signal (2+) and 201-300 = strong signal (3+).”
- The authors are suggested to investigate the underlying mechanisms for statistically significantly correlated proteins. Bioinformatic analysis is not sufficient.
Answer: We also agree that the underlying mechanisms are needed to be established, but that is the whole new study and we leave it for future endeavors.
- Some grammatical errors and typos are present in the manuscript that needs correction. The authors are advised to proofread the paper by a native English speaker and modify the errors.
Answer: We thank you for the advice. We proofread the paper and corrected the errors which is visible through track changes.
Reviewer 2 Report
In the manuscript entitled “Are we benign? What can Wnt signaling pathway and epithelial to mesenchymal transition tell us about intracranial meningioma progression”, Anja Bukovac et al, investigated a total of 72 patients who suffered from meningiomas in order to evaluate the associations of main actors of Wnt signaling pathway and EMT in tumor progression. The study is well structured and the subject is interesting. Although the authors met their goal, some parts of their study need further clarifications.
My points:
- In the international literature there are several original papers and reviews, even by the authors of the present manuscript, that, to my opinion, have added knowledge and understanding of the contribution of major signaling pathways and biological processes to the formation, growth and behavior of meningiomas. Given that in the current study the authors themselves mention in some parts of the text that some of the presented data have already been reported, statement about the novel or innovative aspect to this work should be clarified.
- This study was performed in tumor tissue samples from 72 patients with meningiomas. In the text but also in Table 2., a large number of subgroup analysis is presented and I think that in some instances sample sizes for this analysis may be small enough to render results questionable. Of course, this is not an uncommon practice but often raises the question about the appropriateness of the statistical analyses. When some comparisons are carried out in the presence of very small subsets, this should be noted as a limitation.
- How many of the studied cases were WHO grade II and III tumors? (figures are not so informative for tumor subgroupings). Were any relationships of the studied molecules with recurrence rates or survival?
- In lines 620-621 the authors state: “…although histologically classified as benign, may hide an invasive character”. What they mean? Of the meningioma subtypes of WHO classification, most have a benign course although some distinct variants are more likely to recur or to show aggressiveness; however, they are categorized as WHO grade II and III.
- In line 563, the expression “…histological type of grade” does not seem correct since histological typing is different from histological grading.
- Immunohistochemical figures of better quality, if possible, would be desirable.
- Please add in the section of “Materials and Methods-Tissue collection” the ethical approval number of the study.
- The manuscript is well written and very few grammar errors are noticed, eg. Line 15: meningiomas instead of meningioma; line 99 investigate instead of investigates.
Author Response
Comments of Reviewer #2:
Rviewer 2 stated that the study was well structured and the subject was interesting. Although the authors met their goal, some parts of their study needed further clarifications.
- In the international literature there are several original papers and reviews, even by the authors of the present manuscript, that, to my opinion, have added knowledge and understanding of the contribution of major signaling pathways and biological processes to the formation, growth and behavior of meningiomas. Given that in the current study the authors themselves mention in some parts of the text that some of the presented data have already been reported, statement about the novel or innovative aspect to this work should be clarified.
Answer: Our study brings first thorough investigation of the interplay between EMT and Wnt signaling pathway in meningioma. To our best knowledge, cadherin switch was never investigated in terms of intracranial meningioma. Although there are few papers that investigate some of molecules presented in our article, the selection of key players of Wnt signaling pathway and EMT were never put in correlation. The novelty of the present study is the role and associations relevant molecules play in meningioma grades and types. To clarify this in our article we have rewritten the conclusions as it follows:
“This study sheds light and brings novel insights on the role of important molecules of the Wnt signaling pathway and EMT in the development and progression of intracranial meningioma. We showed that genes involved in cadherin switch – CDH1 and CDH2 may play a role in the development and progression of meningiomas, where both genes are involved and the basis for defective EMT has been established. Protein analyses also revealed weaker expression of E-cadherin in relationship to N-cadherin, suggesting EMT in meningioma. Our investigation on EMT transcriptional factors TWIST1, SNAIL and SLUG demonstrated their strong expression, where SNAIL and SLUG were associated to higher grades and represent a good indicator of the progression of intracranial meningiomas. Results on β-catenin showed that it is mostly present in its active form. Also, active form of β-catenin was significantly associated to higher grades suggesting the role of the Wnt signaling pathway as a driver of meningioma progression. Our protein analyses of E-cadherin, N-cadherin, SNAIL, SLUG, and TWIST1 and two forms of β-catenin demonstrated a close association between Wnt signaling pathway activation and epithelial-mesenchymal transition in intracranial meningiomas.”
- This study was performed in tumor tissue samples from 72 patients with meningiomas. In the text but also in Table 2., a large number of subgroup analysis is presented and I think that in some instances sample sizes for this analysis may be small enough to render results questionable. Of course, this is not an uncommon practice but often raises the question about the appropriateness of the statistical analyses. When some comparisons are carried out in the presence of very small subsets, this should be noted as a limitation.
Answer: Having a small sample sizes is not necessary limiting if right statistical analysis is conducted. Our statistical analysis was performed as it follows: Before selecting appropriate statistical analysis for data processing, the distribution normality of the included variables was tested by the Kolmogorov-Smirnov test. Whenever possible, parametric tests were used for data analysis, which in principle have a higher statistical power and are more discriminative thus you are more likely to detect a significant effect when one truly exists. In the event that the variables showed a statistically significant deviation from the normal distribution, the obtained conclusions were further verified by an equivalent nonparametric procedure. If both test confirmed statistical significance, the result was considered valid. Regardless of the number of samples, statistical analysis showed us there was a significant difference between subgroups, even in those with small number of samples, where is more difficult to obtain significance, which proves the importance of this findings and should be further explored.
- How many of the studied cases were WHO grade II and III tumors? (figures are not so informative for tumor subgroupings). Were any relationships of the studied molecules with recurrence rates or survival?
Answer: We agree that it was not clearly stated how many cases of each grade we have collected. Therefore, we have added to the Materials and methods section Tissue collection on page 3, line 109 the following sentence: “Our sample consisted of 49 benign, 17 atypical and 6 anaplastic meningiomas”.
Relationship of the studied molecules was not examined with recurrence or survival rate due to insufficient information. Unfortunately, we were able to collect patient survival for only 34 out of 72 patients. Information for other patients were unavailable. The next problem is that out of 34 patients, only 3 have passed the 5-year survival rate and 28 are still alive but their 5-year survival rate still has not run out (and hopefully will not). Also, some patients never returned for follow-up and the medical personnel does not have the information if it is because of patients’ death or due to getting treatment in local hospital (patient data and findings are not networked in the state). We know for sure that one patient with grade III tumor died, while two patients with tumors II and III are speculated to be dead after not returning for follow-up. Still for those two patients we cannot be sure if death was due to their tumor state or maybe some other illness. Therefore, any statistical correlations with this partial data would be insufficient and would not present the real situation.
- In lines 620-621 the authors state: “…although histologically classified as benign, may hide an invasive character”. What they mean? Of the meningioma subtypes of WHO classification, most have a benign course although some distinct variants are more likely to recur or to show aggressiveness; however, they are categorized as WHO grade II and III.
Answer: It is true that main pathological classification of meningioma divides them into three categories according to their invasiveness, with benign grade being the most benevolent. Still, the new research and classification of tumors showed that tumors of the same pathohistological type do not always have the same behavior or response to therapies based on differences in molecular pathways or genetic profiles regardless of grade. Today, the molecular profile of tumors has become crucial in the diagnosis and choice of therapy, and the WHO confirmed it in its 2016 classification (Louis et al., 2016 https://pubmed.ncbi.nlm.nih.gov/27157931/). For instance, novel studies showed that meningiomas can be divided into six methylation categories that have been described to be better predictors of recurrence than pathohistological classifications and grade, especially for grade II meningiomas. The first three categories determine the benign course of disease development, while the other three show a poorer prognosis and outcome of the disease (the last category is related to aggressive tumors). Thus, benign meningiomas may belong to a particular methylation group that will show similarities to higher-grade meningiomas behavior and will be more likely to recur than benign meningiomas that do not belong to that methylation group (Buerki et al., 2018 https://pubmed.ncbi.nlm.nih.gov/30084265/; Scheie et al., 2019 https://pubmed.ncbi.nlm.nih.gov/30740783/).
To state this more clearly, we added explanation to the sentence: “The results of this study are consistent with this claim since we also observed strong expression of SNAIL, SLUG, and TWIST1 in some cases of benign meningioma, reaffirming that meningioma, although histologically classified as benign, may hide an invasive character due to their molecular landscape or activated pathways and driving mutations, which can lead to their progression.”
- In line 563, the expression “…histological type of grade” does not seem correct since histological typing is different from histological grading.
Answer: We agree with the statement and apologize for the mistake. The sentence was corrected:” Because of all of the above, N-cadherin could potentially be a good prognostic marker of meningioma behavior regardless of histological grading.”
- Immunohistochemical figures of better quality, if possible, would be desirable.
Answer: We have added new figures taken on our new light microscope. The figures are in new, bigger size in order to conserve the quality.
- Please add in the section of “Materials and Methods-Tissue collection” the ethical approval number of the study.
Answer: We added the ethical approval numbers of the study to the section of “Materials and Methods-Tissue collection” on page 3, lines 111-116. The ethical approval is also stated in the section Institutional Review Board Statement: The study was conducted according to the guidelines of the Declaration of Helsinki, and approved by Ethics Committee of School of Medicine Universi-ty of Zagreb (Case number: 380-59-10106-17-100/98; Class: 641-01/17-02/01, 23. 03. 2017), Ethics Committee of University Hospital Center Zagreb (number 02/21/AG, class: 8.1-16/215-2, 02. 02. 2017), Ethics Committee of University Hospital Center “Sisters of Charity” (number EP-5429/17-5, 23. 03. 2017) and Ethics Committee of University hospital Dubrava, (17.05.2017).
- The manuscript is well written and very few grammar errors are noticed, eg. Line 15: meningiomas instead of meningioma; line 99 investigate instead of investigates.
Answer: We thank you for the observation and advice. The errors were corrected in the article.

Reviewer 3 Report
The study looks interesting. The method is valid.
Authors should better describe the correlations between their data and the histological grade of meningiomas.
Furthermore, if possible, it would be interesting to report any correlations of the data obtained, follow-up and patient survival.
Author Response
Comments of Reviewer #3:
Rviewer 3 stated that the study looked interesting and the method was valid. However, authors should address following:
- Authors should better describe the correlations between their data and the histological grade of meningiomas.
Answer: We agree that significant correlations were not clearly described so we rewrote sentences regarding this issue:
- In chapter The role of SNAIL, SLUG and TWIST1 transcription factors in intracranial meningioma: “Further analysis showed a statistically significant association of SNAIL and SLUG protein expression with different grades (p = 0.001), which was also true for protein expression in nuclei (p = 0.001) suggesting its accumulation and role in higher grades. This result is consistent with other studies that also report that the expression of SNAIL and SLUG in nuclei was associated with higher malignant potential and poorer prognosis in other tumors such as malignant pleural mesothelioma and in lung neuroendocrine tumors [63,64].”
- In subsection Role of β-catenin phosphorylation status and Wnt signaling pathway in intracranial meningioma: “The results of active β-catenin expression in the grade II and grade III samples of this study suggest that the Wnt signaling pathway plays an important role in meningioma and their progression.”
We also added a paragraph in conclusion: ” This study sheds light and brings novel insights on the role of important molecules of the Wnt signaling pathway and EMT in the development and progression of intracranial meningioma. We showed that genes involved in cadherin switch – CDH1 and CDH2 may play a role in the development and progression of meningiomas, where both genes are involved and the basis for defective EMT has been established. Protein analyses also revealed weaker expression of E-cadherin in relationship to N-cadherin, suggesting EMT in meningioma. Our investigation on EMT transcriptional factors TWIST1, SNAIL and SLUG demonstrated their strong expression, where SNAIL and SLUG were associated to higher grades and represent a good indicator of the progression of intracranial meningiomas. Results on β-catenin showed that it is mostly present in its active form. Also, active form of β-catenin was significantly associated to higher grades suggesting the role of the Wnt signaling pathway as a driver of meningioma progression. Our protein analyses of E-cadherin, N-cadherin, SNAIL, SLUG, and TWIST1 and two forms of β-catenin demonstrated a close association between Wnt signaling pathway activation and epithelial-mesenchymal transition in intracranial meningiomas.”
- Furthermore, if possible, it would be interesting to report any correlations of the data obtained, follow-up and patient survival.
Answer: We agree that follow-up and patient survival are of great importance for this type of studies. Unfortunately, we were able to collect patient survival for only 34 out of 72 patients. Information for other patients were unavailable. The next problem is that out of 34 patients, only 3 have passed the 5-year survival rate and 28 are still alive but their 5-year survival rate still has not run out (and hopefully will not). Also, some patients never returned for follow-up and the medical personnel does not have the information if it is because of patients’ death or due to getting treatment in local hospital (patient data and findings are not networked in the state). We know for sure that one patient with grade III tumor died, while two patients with tumors II and III are speculated to be dead after not returning for follow-up. Still for those two patients we cannot be sure if death was due to their tumor state or maybe some other illness. Therefore, any statistical correlations with this partial data would be insufficient and would not present the real situation.

Round 2
Reviewer 1 Report
The authors revised the manuscript in response to all comments from the Reviewer. They detail their responses and revisions made point-by-point and the Reviewer feels that the manuscript is much improved and hope it will be acceptable for publication in the journal.
Reviewer 2 Report
I re-reviewed the attached article entitled “Are we benign? What can Wnt signaling pathway and epithelial to mesenchymal transition tell us about intracranial meningioma progression” by Anja Bukovac et al.
I have seen changes in this manuscript since the first version. The authors responded to the reviewer’ s comments and they restructured or clarified some sections of the article. To my opinion, presented data is a good starting point for further study.
I consider the manuscript suitable for publication in “Cancers”.